# A Digital Twin-Based Distributed Manufacturing Execution System for Industry 4.0 with AI-Powered On-The-Fly Replanning Capabilities

Jiří Vyskočil [1,*,†], Petr Douda [1,†], Petr Novák [1,†] and Bernhard Wally [2]

1   Czech Institute of Informatics, Robotics and Cybernetics, Czech Technical University in Prague, Jugoslávských Partyzánů 1580/3, Dejvice, 160 00 Prague 6, Czech Republic
2   Office of the Austrian Council for Research and Technology Development, Pestalozzigasse 4/D1, 1010 Vienna, Austria
*   Correspondence: jiri.vyskocil@cvut.cz
†   These authors contributed equally to this work.

**Abstract:** Industry 4.0 smart production systems comprise industrial systems and subsystems that need to be integrated in such a way that they are able to support high modularity and reconfigurability of all system components. In today's industrial production, manufacturing execution systems (MESs) and supervisory control and data acquisition (SCADA) systems are typically in charge of orchestrating and monitoring automated production processes. This article explicates an MES architecture that is capable of autonomously composing, verifying, interpreting, and executing production plans using digital twins and symbolic planning methods. To support more efficient production, the proposed solution assumes that the manufacturing process can be started with an initial production plan that may be relatively inefficient but quickly found by an AI. While executing this initial plan, the AI searches for more efficient alternatives and forwards better solutions to the proposed MES, which is able to seamlessly switch between the currently executed plan and the new plan, even during production. Further, this on-the-fly replanning capability is also applicable when newly identified production circumstances/objectives appear, such as a malfunctioning robot, material shortage, or a last-minute change to a customizable product. Another feature of the proposed MES solution is its distributed operation with multiple instances. Each instance can interpret its part of the production plan, dedicated to a location within the entire production site. All of these MES instances are continuously synchronized, and the actual global or partial (i.e., from the instance perspective) progress of the production is handled in real-time within one common digital twin. This article presents three main contributions: (i) an execution system that is capable of switching seamlessly between an original and a subsequently introduced alternative production plan, (ii) on-the-fly AI-powered planning and replanning of industrial production integrated into a digital twin, and (iii) a distributed MES, which allows for running multiple instances that may depend on topology or specific conditions of a real production plant. All of these outcomes are demonstrated and validated on a use-case utilizing an Industry 4.0 testbed, which is equipped with an automated transport system and several industrial robots. While our solution is tested on a lab-sized production system, the technological base is prepared to be scaled up to larger systems.

**Keywords:** production system; Industry 4.0; digital twin; AI planning; simulation; flexible automation system; robotics; manufacturing execution system

## 1. Introduction

Energy-wise sustainability of today's industrial production systems [1] is a major challenge (available online: https://www.senseye.io/hubfs/Downloads/Senseye-Challenges-of-Sustainability.pdf, accessed on 11 January 2023), both in terms of (i) the use of more

efficient production technologies and (ii) new algorithms for their automation and control [2,3]. Software development and maintenance of such systems account for a significant part of their acquisition and operating costs [4,5]. Consequently, according to our extensive experience from many industrial projects, control systems of a large proportion of today's industrial systems are programmed in a single-purpose manner with the aim of meeting high operational reliability and repeatability. Other parameters of these control systems, such as energy savings during production restarts [6], are not considered at all or only to a very limited extent. This is mainly due to the unpreparedness of traditional industrial engineering and programming practices for rapid development [7], validation, and debugging, and the absence of an advanced and flexible infrastructure that can perform some parts of production optimization fully or at least partially automatically.

Smart manufacturing [8,9], or Industry 4.0 (I40) [10], includes the integration of cyber-physical systems [11], their mutual interconnection using universal communication protocols with the transfer of semantic information, the integration of artificial intelligence (AI) in planning, scheduling [12,13], and optimizing production [14], digital twins [15], the internet of things (IoT), and more [16,17]. Current trends, as well as already achieved I40 goals, are described in a recent paper by Kagemann and Wahlster [18]. The role of Industry 4.0 in the domain of sustainable industrial production is addressed in [19–21].

Contemporary production systems are typically characterized by a hierarchically layered architecture, often referred to as *automation hierarchy* [22]. At its lowest level (closest to the equipment), the *process data inputs and outputs (I/Os)* provide direct connection to all industrial components such as robots, sensors, transportation systems, etc. One layer above, the *programmable logic controllers (PLCs)* run low-level control algorithms for basic functions, thus operating the interfaces among shop-floor devices, and assuring the safety features related to specific parts of the automation system. Next is the *supervisory control and data acquisition (SCADA)* [23] layer, that implements a control system architecture including processing units, data communication, and graphical user interfaces for high-level supervision of machines and processes. The *manufacturing operations management (MOM)* [24] layer is located on top of SCADA and functions as a real-time system that allows for the controlling of multiple elements of the production line according to the current production plan by utilizing a *manufacturing execution system (MES)*. On the top vertex of the automation stack lies an *enterprise resource planning (ERP)* [25] layer, which is responsible for the integrated management of core business processes and usually also provides longer-term and higher-level production and resource planning.

With the reference architecture model of Industry 4.0 (RAMI40) [26], a more timely view on the smart manufacturing domain is provided, featuring a three-dimensional holistic map [27]. textcolorredMDPI: Please check that intended meaning is retained.However, RAMI40 is not an implementation approach, nor does it prescribe specific requirements that can be directly checked while provisioning production systems. As such, the trajectories towards I40 manufacturing systems are still very challenging to achieve [28].

This article describes a comprehensive concept for the I40 production line testbed located at the Czech Technical University (CTU) in Prague—Czech Institute of Informatics, Robotics and Cybernetics (CIIRC). The entire production line has its own continuously synchronized digital twin with a formal symbolic representation of all feasible production operations in each line state. This digital twin is used as a source of semantic information for fully automated AI-powered production planning realized using the planning domain definition language (PDDL) [29]. Thanks to this architecture, which will be described in detail in Section 3, it is possible to on-the-fly inject new production goals or automatically improve existing plans for current production goals according to actual changes and needs, and therefore seamlessly alter the ongoing production process.

The distributed MES that we have implemented in this work allows the execution of the production plan to be split among hierarchically interconnected and synchronized MESs. With this decomposition, each individual MES can fully control all of its subordinate industrial components in real-time with low latency, which are assumed to be interconnected using OPC

Unified Architecture (UA) [30]. However, for interconnection of individual MESs among each other, we have implemented a representational state transfer (REST) application programming interface (API) approach, given that OPC UA is not very suitable for unreliable communication with high latency, as shown in the performance evaluation study [31] of different communication protocols for I40. In contrast, our RESTful approach allows the MESs to be synchronized over much larger distances (e.g., across buildings, cities, even countries) where network characteristics are considered to feature lower bandwidth, lower connection reliability, and higher latency as compared to an in-house production line network.

Compared to previous work [32], the distributed MES presented in this article is extended with the ability to change the production plan on-the-fly. Similar to the approach described in an earlier article [33] our implementation allows

(i)     for a continuous search for a better and better production plan, and
(ii)    the seamless conversion of an ongoing production to such a newly found plan without requiring a global and time-consuming restart of the entire production process.

Compared to that earlier article and further previous work [34], we have integrated into this work the concept of a *distributed* MES with replanning capabilities.

The remainder of this article is structured as follows. Section 2 presents related work in terms of providing a background for understanding our improvements and achievements. Section 3 explains the proposed solution involving a distributed MES that utilizes a digital twin enhanced with on-the-fly AI-powered planning, replanning [35,36], and scheduling capabilities/features. Section 4 demonstrates the proposed solution on a real example and discusses the properties of results achieved. Section 5 concludes and suggests future work.

## 2. Materials and Methods

The most fundamental contribution of this article is the improved planning and scheduling of industrial production systems that are controlled by a distributed MES. Therefore, the state-of-the-art presented in this section provides a background and foundation for describing and understanding the proposed contribution.

### 2.1. AI Planning and Scheduling

Today, advanced planning using artificial intelligence (AI) is receiving more and more attention in order to ensure high flexibility and efficiency in industrial production processes. Flexibility means that an ongoing production can automatically respond to various situations without manual intervention in the production control source code. Efficiency means, for example, that the production process is faster, less expensive, or more energy efficient; or it can combine several of these factors. It is also important to note that the terms *planning* and *scheduling* are frequently applied in a variety of contexts with diverse meanings.

Automated planning (also called AI planning [37]) is an area of AI that solves the problem of finding a sequence of operations that must be completed to achieve a given goal given a certain starting condition and a set of possible actions. This sequence is called a *plan*. The set of possible actions, including their pre- and post-conditions, as well as the available relations between objects, is expressed in a *domain* description. The current state (the starting condition) of all participating objects and their relations, as well as the aspired state (the goal condition), are stated in a *problem* definition. Problems and domains can be specified in a special language developed for AI planning, the planning domain definition language (PDDL), which was introduced by McDermott et al. in 1998 [38]. Meanwhile, PDDL has reached maturity (its latest version (3.1) was released in 2011 [39]), even being deployed in industrial environments. Over time, various extensions to PDDL have been developed, supporting, for example, modal operators, ontologies, probabilistic effects, partial observability, goal rewards, durative actions, hierarchical action expansion, and many more. Applications of some of these PDDL extensions in the context of production and logistics, including a description of the techniques used in the solvers, are discussed by Sousa and Tavares [40], while durative actions have, e.g., been utilized by Wally et al. [41].

Scheduling (as described by Pinedo [42]) refers to the problem of finding optimal or sub-optimal schedules (such as Gantt charts) for executing finite (or repetitive) sets of tasks/jobs, often related to resource capacities or soft and hard constraints among tasks and resources. The problems addressed by scheduling can be formalized as optimization problems to process a finite set of tasks in a system with limited and constrained resources. In the scheduling, the time of arrival is specified for each task. In the production system, each task passes through multiple processing phases that depend on the input conditions of the problem. For each phase, feasible resource sets are assigned, and processing times according to the selected resources are considered.

### 2.2. Planning with PDDL

Planning problems that are expressed using PDDL differentiate between two separate specification files that describe (i) the domain of the problem and (ii) a specific problem instance within this domain, as follows (the underlying PDDL domain and problem descriptions used in this article are depicted in Figure 1):

**Figure 1.** A simplified PDDL domain specification (**left**) and an example problem (**right**) for the Montrac transportation system, PDDL keywords are highlighted in blue. The domain specification lists the available types, predicates, and actions (along with their parameters, preconditions, and effects). The problem specification encodes (i) the current state of the Montrac system, that is, which shuttles are in operation, where are they located, what is the topology of connections, and (ii) what the current production goal to be achieved is. The problem specification is regularly updated by the digital twin of the production line (Figure 2).

**Domain:** specification of the *types* of entities that are available in this domain, their *predicates* and *functions* as well as all the available *actions* including their input *parameters*, their *preconditions* that must hold before a specific action begins, and their *effects*, which are changes in the state-space done immediately after a specific action is finalized. Effects can optionally have assigned advanced attributes such as costs/fitness and durations. Further, the used language extensions need to be specified in terms of *requirements*.

**Problem:** specifies a particular instance of the problem, which contains a description of the *initial* state, i.e., the available object instances and their properties and relations to

another (expressed through predicates) and the definition of the aspired *goal* state, i.e., the predicate expression that needs to be evaluated in boolean `true`.

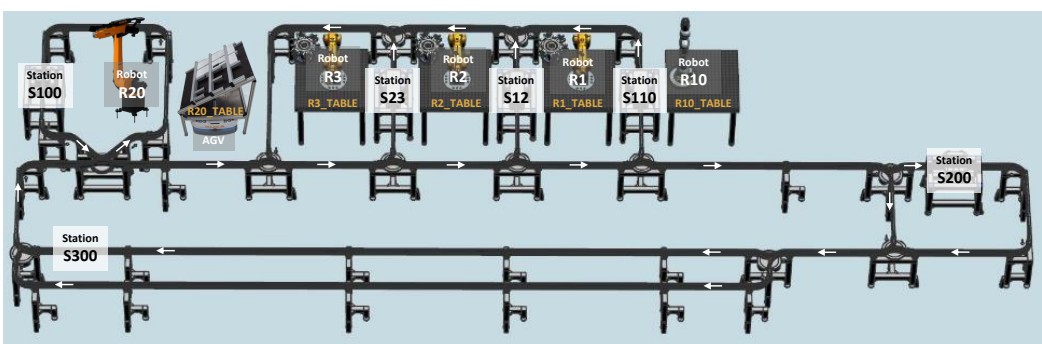

**Figure 2.** Topology of the I40 testbed's flexible production line. It is equipped with five robots, one AGV for manipulation with $R20\_TABLE$, a transportation system operated by six shuttles, interconnecting six workcells/workstations. The shuttles are not included in the figure because they can have any actual position according to the current needs. Workstations $S23$, $S12$, and $S110$ are accessed by robots located on both sides; and the workstation $S100$ is reachable only by robot $R20$.

The solution of a planning problem specified in PDDL is called a *plan*, i.e., it is a sequence of actions that have to be executed when beginning at the initial state of the planning problem, with the goal condition being satisfied after all actions have been processed.

### 2.3. Digital Twins for Industrial Systems

A *digital twin* can be described as a digital replica (mathematical abstraction) of a physical system. The digital twin concept consists of three parts: (i) the physical entity (object or process) and its physical environment, (ii) the digital representation of the entity, and (iii) the communication channel between the physical and virtual representations. The bi-directional link between the physical version/twin and the digital version/twin involves information flows and data, which include physical sensor flows between the physical and virtual objects and environments.

The first concept of digital twin was published by Grieves in 2002 (available online: https://www.researchgate.net/publication/307509727_Origins_of_the_Digital_Twin_Concept (accessed on 11 January 2023), and since then, the concept has been widely used in the context of I40 [43–46].

Digital twins are usually designed domain-specifically, relying on a mathematical-physical description of physical phenomena and their parameterization (including properties like shapes, materials, or others). Papers by Hänel et al. [47,48] are good examples of mathematical-physical descriptions, in this case of machining processes for high-tech/aerospace industry, of domain-specific digital twins. Digital twins can also be automatically or semi-automatically generated from various existing information sources. An example of such creation of digital twins coming out of 2D and 3D CAD models is discussed by Sierla et al. [49]. It generates graphs from available models, which are used for graph matching algorithms. Furthermore, improvements to this solution were investigated in the follow-up paper [50], targeting the generation of digital twins for brownfield process plants.

An explanation of the difference between *simulation* and *digital twin* was given by Kritzinger et al. [51]. According to Kritzinger et al., digital system approximations fall into three categories according to the level of integration: (i) *digital models*–a stand-alone execution independent from the data from physical artifacts or with manual data exchange only, (ii) *digital shadows*–updated with automated one-directional data flow from physical artifacts to their digital counter-parts, and (iii) *digital twins*–synchronized with bi-directional data flows among physical artifacts and their digital counter-parts. That paper includes a systematic literature review of other digitalization solutions as well. Only a small part of them are categorized as *digital twins* according to this aforementioned difference.

The solution presented in this article utilizes the bi-directional data flows among digital and physical artifacts, and therefore the solution can be classified into the category of digital twins (iii).

### 2.4. Manufacturing Execution Systems

A systematic review on present trends in research and development in the MES area is provided by Shojaeinasab et al. [52]. The limited capabilities of legacy commercial MESs are discussed by Bratukhin and Sauter [53]. Despite the fact that the reference is already one decade old, the offer on the market has not improved sufficiently in the meantime. The paper spot-lights a set of common corner-stones of a typical distributed MES [54], which are *order managers*, *resource managers*, *supervisors*, and *brokers*. Such a distinction goes along with the contribution by Mařík and McFarlane [55] that deals with threats and strengths of adopting a multi-agent design principle for industrial automation and control, particularly at the level of manufacturing execution.

The structure and implementation of generic MESs are addressed by Fei [56], by identifying the fundamental components of MES: *equipment management*, *production process management*, *quality management*, *order management*, *production scheduling management*, and *resource management*.

Various achievements in the area of manufacturing execution are discussed by Pan et al. [57]. These emerging trends cover: *cloud-based MES*, *IoT-based MES*, *intelligent MES*, *collaborative MES*, *supply chain linkage*, *MES mobility*, and *industrial data analysis*. A special demonstration of an IoT-based MES implemented for a motor plant is given by Gao et al. [58], where a wide range of sensor data could be accessed through IoT means.

The conceptual integration between the ERP and MES layers is standardized by ISA-95 [59], which has since become an international standard (IEC 62264) [22]. A concrete alignment, including potential mapping issues, between an ERP ontology and ISA-95 has been presented by Wally et al. [60], a graphical toolkit for manipulating ISA-95 models by Lang et al. [61]. For the utilization of ISA-95 from within AutomationML model,s it is useful to follow the corresponding application recommendation [62]. AutomationML is a standardized data exchange format for I40 engineering tools [63], and is used in our approach as described in Section 3. The integration between MES and shop-floor devices (that is, between the two bottom-most levels of the automation pyramid) has already been standardized by many communication protocols. In emerging I40 system architectures, the most frequent protocol is OPC UA [64]. It is an open-source, cross-platform industrial standard (IEC62541) that unifies data representation, access, historical data access, and alarms and events into a single and coherent specification [30]. OPC UA can be utilized for control and data acquisition from automation devices, including smart sensors, programmable logic controllers (PLCs), or robot controllers. One of the most significant advantages of OPC UA (in comparison to most of the legacy industrial communication protocols) is that it is not restricted for the classic client-server communication only, but also supports publish-subscribe communication, causing the communication to be more efficient by eliminating exhaustive and unnecessary/redundant polling. Due to its versatility and flexibility, the OPC UA protocol is thus the preferred communication protocol in I40 environments. It can be used for a flexible plug and produce system architecture and integration [65,66]. Support for ISA-95 within OPC UA has been standardized by a *companion specification* [67]. Further information about the use of standards, together with formal modeling approaches in I40 production systems is given in a dedicated book chapter [68].

### 2.5. Industry 4.0 Smart Manufacturing Enabled by PDDL and Digital Twins

Manufacturing processes in automated manufacturing systems need to be capable of being incrementally updated, modified, and evolved throughout the entire life cycle of the manufacturing system. Consequently, this means, among other things, that software components and their source codes need to evolve accordingly, as stated by Vogel-Heuser et al. [69]. The automated generation of simulation models for control code testing has been addressed

by Barth and Fay [70], but this is only a small piece in the overall mosaic of activities in the design process of production system engineering.

The use of PDDL specifications in the planning of industrial problems is discussed by Rogalla et al. [71]. They provide a collection of generic prototype cases, including the corresponding formal representations. In contrast to that paper, the approach proposed in this article targets more complex industrial-scale problems/systems (an industrial-scale system is defined as a system whose topology includes such number and types of devices that are comparable to a real environment of a production system in industry. In this article, the use-case (see Section 3.1) comprises five robots arranged in a structure of six workstations that are called by a transportation system with six shuttles. Communication is implemented using OPC UA). Further, we use the PDDL specification not only for offline planning, but also for on-the-fly planning, that is tightly integrated with the digital twin. In other words, the solution proposed in this article poses an integrated intelligent production planning and execution system that can react immediately to any change identified in the production line or production goals.

The approach proposed by this article is based on our systematic investigation and research in the field of advanced AI-based planning using PDDL. The production line is symbolically represented in PDDL and it is bi-directionally synchronized with our proposed distributed MES. An early proposal of a goal-oriented manufacturing execution using the PDDL specification for automated planning is addressed in our paper [72] from 2019. It already contains the concept of PDDL-based symbolic digital twin integration, however, this original, but preliminary, concept is not described in detail there. The complete concept of PDDL as a digital twin for I40 smart manufacturing was described in a follow-up paper [73], and thoroughly evaluated by a further contribution [33].

An automatic translation of formal models for production system engineering into PDDL representations is described in one of our initial contributions [74]. There, a set of rules was proposed to directly convert a model for the automated production system into a planning domain specification and a planning problem. The experiments performed there have shown that even fully automated planning (demonstrated on a single scalable use-case pattern/template) is feasible if the available models provide sufficient information. Specifically, it allowed for the direct use of the vendor's configuration file with a topology description of the Montrac transport system. The main idea behind this approach was based on concepts stemming from model-driven software engineering, enabling a formal expression of domain metamodels and concrete instances that can be used for automated transformation between different metamodels [75].

Subsequently, this approach was altered in the next paper [41] with the *durative actions* extension of PDDL. This capability has moved the implemented production planning, where the plan is a sequence of actions, more into the realm of production scheduling, where a parallel run of actions is allowed. On the one hand, solutions of such planning/scheduling problems can be of very high quality in terms of total duration (total production time). The reason is rooted in paralleling the concurrent actions/operations (considered with their exact time-durations) on multiple production resources precisely inside the dedicated PDDL planner. On the other hand, the planning with durative actions leads to increasing the computational complexity of planning enormously, with regard to the size of the domain and the problem. It has been shown in [41] that this PDDL extension is rather suitable only for problems of small size—for problems of similar size to our Industry 4.0 testbed, the time required for planning/scheduling is already prohibitively long and also memory intensive. Based on this experience, in this article, we do not use durative actions and instead split planning and scheduling into two separate processes.

## 3. Implementation and Results

This section presents the proposed solution in detail. Prior to describing the new system architecture and the whole solution, the following description addresses the testbed environment, which was used for testing, validating, and fine-tuning of the proposed

solution. Moreover, it provided us with the initial motivation and ideas for this research direction.

### 3.1. Industry 4.0 Testbed

CIIRC's Industry 4.0 testbed represents a unique environment to foster the technology and knowledge transfer from science and academic research to industrial engineering practice. The main mission is to make I40 visions real through concrete implementations. Achievements and shifts in industrial production practice are presented to a wide audience including students, engineers, experts, and other enthusiasts. This high-tech facility is suitable for exploring new ways of control, optimizing the operation of robotic production lines, innovating and designing new robotic cells, and supporting industrial control in conjunction with energy consumption and optimization.

The entire research laboratory is accommodated to provide energy analyses and optimizations. The laboratory includes power line connections to renewable energy sources, namely photovoltaic panels located on the roof of the building. The photovoltaic system is accompanied by additional battery storage, enabling smart load balancing and smoothing of the energy demand from the distribution grid. All robots are equipped with power and energy meters that allow continuous monitoring of power and energy consumption. In the area of energy sustainability, the I40 testbed is an important body for analyzing and evaluating the energy needs of medium- and large-scale robotic systems. It provides useful support to industrial companies by recommending suitable solutions and as a physical testbed for evaluating optimization scenarios.

In this article, we focus on a so-called flexible production line, which is a core system of the I40 testbed. It consists of five industrial robots of three types: (i) a collaborative robot, the KUKA LBR iiwa (frequently called cobot); (ii) three small and fast industrial robots, the KUKA Agilus; and (iii) one extended-range robot, the KUKA Cybertech (cf. Figure 3). Material is brought to the production system by automated guided vehicles (AGVs) for intelligent mobile transport of the type KUKA KMP. These mobile platforms are capable of moving pallets with semi-products on their tops and with a maximum load of 600 kg.

Robotic workcells are interconnected by the Montrac transport system (cf. Figure 2). It is composed of a set of rails and junctions on which shuttles are moving. The rail system is of a mono-rail nature, providing a stable positioning of shuttles. Each shuttle has its own control unit and motor/powerdrive, so the shuttles move independently from each other (they are equipped with an infrared light barrier at the front to ensure that the shuttle stops if the route is blocked by another shuttle). Compared to traditional belt feeders, the Montrac transport control is distributed and suitable for more difficult path routing. On the other hand, the velocity of shuttles is rather low, therefore, the system is more suitable for highly customized production (with diverse material and semi-product routing) than for mass-production.

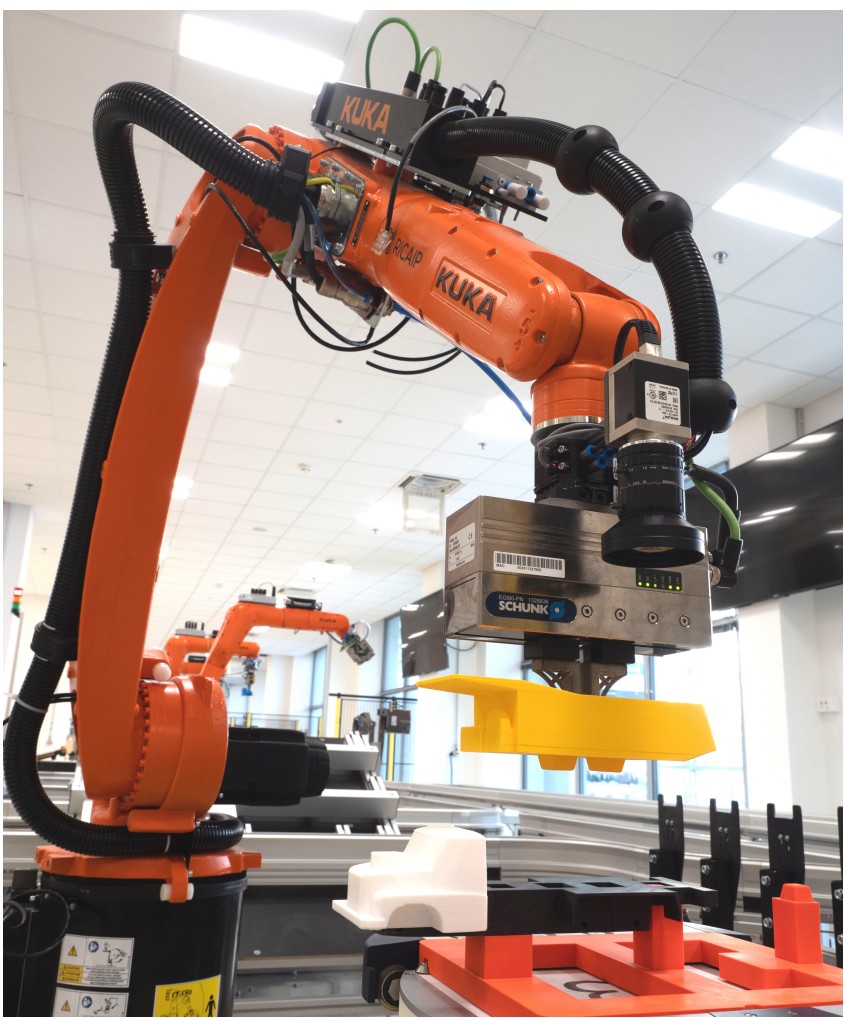

**Figure 3.** The production line of the I40 testbed hosted at the Czech Technical University in Prague with robot *R*20 in the front and robots *R*3, *R*2, and *R*1 in the back. The robot is gripping a yellow car dumper body in its end-effector when completing an assembly process for one of the two trucks from the use-case described later on in Section 4.

From the system formalization perspective, transport and robotic operations are considered "skills". For production planning and scheduling, the available resources are expressed in PDDL, including their supported skills. An important part that has been added is the estimation of energy consumption needed to perform each production operation. In the cost function utilized to specify the optimal production plan, energy needs are one of the factors taken into account.

In our test cases, we focus on the assembly of 3D-printed truck models. Each truck model consists of a chassis as a basic component, which can be further equipped with a cabin, which can have one of four colors, and a body, which can be one of four shapes and one of four colors. There are no fixed production recipes for individual products, but the production recipes are planned by the production planner. During system operation, these generated production plans are executed by a distributed MES by interpreting each production plan operation per operation (some of the operations can be done in parallel on different production resources).

The proposed smart and sustainable industrial production system is enabled by a smart, distributed MES that is capable of interpreting production plans, as described in the following subsection.

### 3.2. Manufacturing Execution System with Dynamically Generated Production Plans

Production orders, which determine what is to be produced, where and when, are key factors in a typical industrial production. These orders are usually first processed by the ERP and then passed on to the MES. The origin of production orders is out of the scope of this article, but we will focus on the format of these orders and the automated technology that is used to process them on a real production line.

Typically, for each production order, a production plan/recipe needs to be newly created by hand or retrieved from a database — it is then executed by the MES on the production line. In our case, the production processes are not hard-coded in the MES, but the production plans are generated dynamically by the planner and scheduler service that is a part of our digital twin (Figure 4). With this architecture, the planner and scheduler have direct access to the current production system state equipped with the continuous synchronization between the digital twin [51,76] and the real production system.

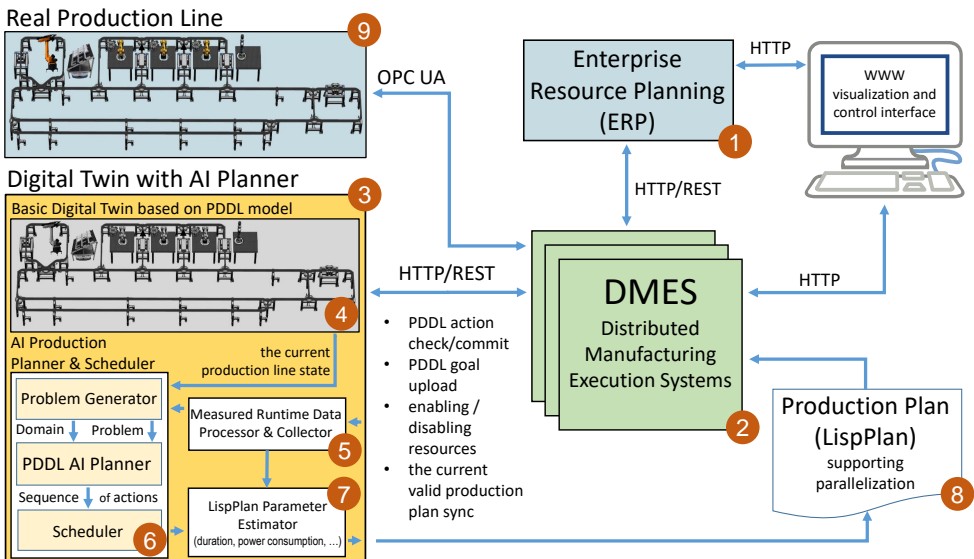

**Figure 4.** Architecture of our automation system, bridging the gap between I40 devices/components on the shop-floor level and the traditional ERP system level. Numbers in circles are used as references for the further textual description of the architecture in details (see Section 3.3).

Our MES simply hands over the production goal in PDDL format to the digital twin and then waits for the corresponding plan to be computed. It then executes the plan, including validation checks for each upcoming action using the digital twin, and also updates the twin's status whenever the real production system changes (i.e., an action has been started, an action has been successfully processed, production resources have changed and as a result the plan has changed, etc.).

The production plan generated by the digital twin is technically a directed acyclic graph (DAG) in Lisp (available online: https://lisp-lang.org/, accessed on 11 January 2023) syntax, which we will denote as *LispPlan*. It includes information about all actions/tasks (DAG vertices), their corresponding parameters, locations (properties of DAG vertices), and requirements/dependencies on other tasks or actions (DAG directed edges). Action vertices, with information about their runtime parameters and unique location, represent I40 components. Directed edges represent dependencies where source vertices specify tasks/actions that have to be executed already before the task/action at the target vertex can commence executing. Tasks can be nested recursively with corresponding sub-tasks (including actions) and arbitrarily (but without loops) interconnected. Figures 5 and 6 show a graphical and a textual representation of such a *LispPlan*, respectively.

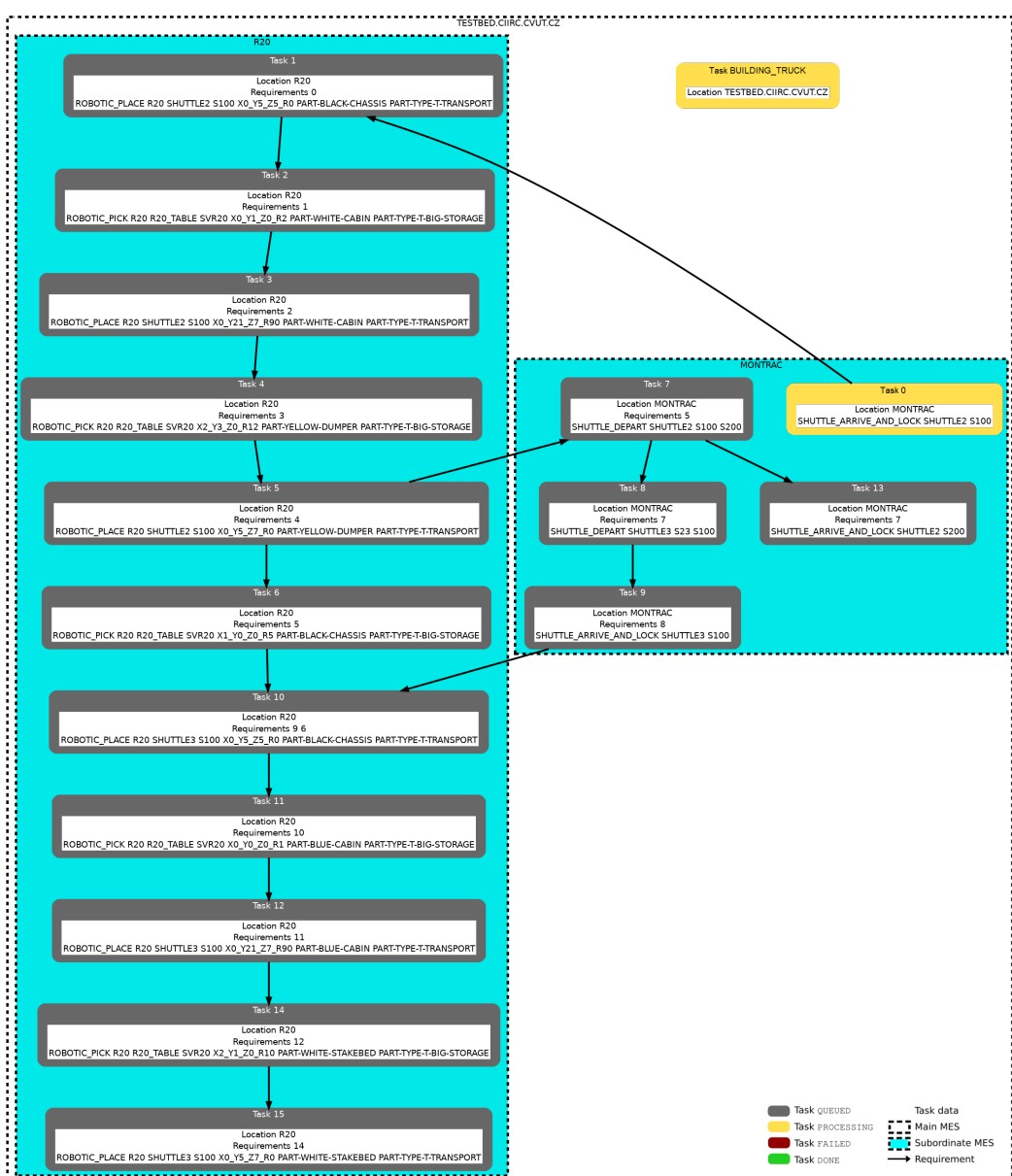

**Figure 5.** Example for a *LispPlan* that was computed in *7s* using one of the faster heuristics (search parameters: `--search "lazy_wastar([ff(),cea(),hmax()], bound=100, boost=0)"`).

### 3.3. Distributed Manufacturing Execution System with On-The-Fly Replanning Capability

Since the MES presented in this article is a distributed MES (DMES), it enables running one or more instances on different servers/computers (uniquely identified by their URLs) and with different competencies (such as I40 devices and sub-MESs). All DMES instances are specified in one configuration file represented in the PyAutomationML [77] data format. PyAutomationML (available online: https://github.com/CIIRC-ISI/PyAutomationML, accessed on 11 January 2023) is our extension for supporting Python 3 injections in the AutomationML (available online: https://www.automationml.org, accessed on 11 January 2023) data format. This extension makes it possible to efficiently represent the entire production line (including I40 devices, digital twins, relationships of PDDL actions with I40 devices, etc.) in a single configuration file.

```
(
    (DEFINE (TASK BUILDING_TRUCK)                                              (DEFINE (TASK 7)
        (:METADATA                                                                (:REQUIREMENTS 5)
            (:ID                                                                  (:LOCATION MONTRAC)
            03C2CADED518237E30C983DF9837FCC3D9BAEEED407FB636D60621A01390A581       (:ACTION (SHUTTLE_DEPART SHUTTLE2 S100 S200)))
            )                                                                  (DEFINE (TASK 8)
            (:DURATION-IN-SECONDS 359.6869435933933)                              (:REQUIREMENTS 7)
            (:TOTAL-TIME-SPENT-IN-SECONDS 426.5899835256443)                      (:LOCATION MONTRAC)
            (:ENERGY-IN-WATT-HOURS 11.306758909567701)                            (:ACTION (SHUTTLE_DEPART SHUTTLE3 S23 S100)))
            (:TOTAL-ACTIONS 16))                                              (DEFINE (TASK 9)
        (:LOCATION TESTBED.CIIRC.CVUT.CZ)                                         (:REQUIREMENTS 8)
        (DEFINE (TASK 0)                                                          (:LOCATION MONTRAC)
            (:LOCATION MONTRAC)                                                   (:ACTION (SHUTTLE_ARRIVE_AND_LOCK SHUTTLE3 S100)))
            (:ACTION (SHUTTLE_ARRIVE_AND_LOCK SHUTTLE2 S100)))               (DEFINE (TASK 10)
        (DEFINE (TASK 1)                                                          (:REQUIREMENTS 9 6)
            (:REQUIREMENTS 0)                                                     (:LOCATION R20)
            (:LOCATION R20)                                                       (:ACTION (ROBOTIC_PLACE
            (:ACTION (ROBOTIC_PLACE                                                       R20 SHUTTLE3 S100 X0_Y5_Z5_R0
                    R20 SHUTTLE2 S100 X0_Y5_Z5_R0                                          PART-BLACK-CHASSIS PART-TYPE-T-TRANSPORT)))
                    PART-BLACK-CHASSIS PART-TYPE-T-TRANSPORT)))             (DEFINE (TASK 11)
        (DEFINE (TASK 2)                                                          (:REQUIREMENTS 10)
            (:REQUIREMENTS 1)                                                     (:LOCATION R20)
            (:LOCATION R20)                                                       (:ACTION (ROBOTIC_PICK
            (:ACTION (ROBOTIC_PICK                                                        R20 R20_TABLE SVR20 X0_Y0_Z0_R1
                    R20 R20_TABLE SVR20 X0_Y1_Z0_R2                                        PART-BLUE-CABIN PART-TYPE-T-BIG-STORAGE)))
                    PART-WHITE-CABIN PART-TYPE-T-BIG-STORAGE)))             (DEFINE (TASK 12)
        (DEFINE (TASK 3)                                                          (:REQUIREMENTS 11)
            (:REQUIREMENTS 2)                                                     (:LOCATION R20)
            (:LOCATION R20)                                                       (:ACTION (ROBOTIC_PLACE
            (:ACTION (ROBOTIC_PLACE                                                       R20 SHUTTLE3 S100 X0_Y21_Z7_R90
                    R20 SHUTTLE2 S100 X0_Y21_Z7_R90                                       PART-BLUE-CABIN PART-TYPE-T-TRANSPORT)))
                    PART-WHITE-CABIN PART-TYPE-T-TRANSPORT)))               (DEFINE (TASK 13)
        (DEFINE (TASK 4)                                                          (:REQUIREMENTS 7)
            (:REQUIREMENTS 3)                                                     (:LOCATION MONTRAC)
            (:LOCATION R20)                                                       (:ACTION (SHUTTLE_ARRIVE_AND_LOCK SHUTTLE2 S200)))
            (:ACTION (ROBOTIC_PICK                                           (DEFINE (TASK 14)
                    R20 R20_TABLE SVR20 X2_Y3_Z0_R12                                  (:REQUIREMENTS 12)
                    PART-YELLOW-DUMPER PART-TYPE-T-BIG-STORAGE)))                    (:LOCATION R20)
        (DEFINE (TASK 5)                                                          (:ACTION (ROBOTIC_PICK
            (:REQUIREMENTS 4)                                                             R20 R20_TABLE SVR20 X2_Y1_Z0_R10
            (:LOCATION R20)                                                               PART-WHITE-STAKEBED PART-TYPE-T-BIG-STORAGE)))
            (:ACTION (ROBOTIC_PLACE                                          (DEFINE (TASK 15)
                    R20 SHUTTLE2 S100 X0_Y5_Z7_R0                                     (:REQUIREMENTS 14)
                    PART-YELLOW-DUMPER PART-TYPE-T-TRANSPORT)))                      (:LOCATION R20)
        (DEFINE (TASK 6)                                                          (:ACTION (ROBOTIC_PLACE
            (:REQUIREMENTS 5)                                                             R20 SHUTTLE3 S100 X0_Y5_Z7_R0
            (:LOCATION R20)                                                               PART-WHITE-STAKEBED PART-TYPE-T-TRANSPORT)))))
            (:ACTION (ROBOTIC_PICK
                    R20 R20_TABLE SVR20 X1_Y0_Z0_R5
                    PART-BLACK-CHASSIS PART-TYPE-T-BIG-STORAGE)))
```

**Figure 6.** Text representation of the *LispPlan* visualized in Figure 5.

The overall proposed architecture of our DMES, along with its digital twin, is shown in Figure 4. Production orders are obtained from the ERP system (shown as 1) using the HTTP-REST interface from the main DMES instance (shown as 2). The main DMES instance then uploads the production goal in PDDL format to the digital twin (shown as 3) that includes a planning and scheduling service (shown as 6), which combines this goal specification with the PDDL domain formalization of the production line and the current/latest state of the production line, which is retrieved from the encapsulated basic digital twin (shown as 4) containing only the PDDL model. The expected costs of actions for the PDDL AI planner are obtained empirically from the *Measured Runtime Data Processor and Collector* (shown as 5). If a plan is found by the PDDL planner, then this sequential plan is parallelized and translated by the scheduler into a *LispPlan* and then passed to the *LispPlan Parameter Estimator* (shown as 7), which calculates estimates of expected remaining time, energy consumption, etc., and adds these calculations as metadata to the *LispPlan*. Then, the final *LispPlan* (shown as 8) is passed back to the main DMES instance in order to be used for the next concrete production steps.

A typical DMES application consists of several instances of MES, with one of them representing the main instance. The specific number of DMES instances depends on the size of the real system and its spatial distribution. An example of a structure with five DMES instances is shown in Figure 7. The main instance is denoted by $MES_0$. This main instance is the only instance that is able to receive the whole *LispPlan* and then delegates the respective parts of this *LispPlan* directly to subordinate MESs. The "direct subordination" relationship is fully specified by the location parameter of each DMES instance: just like the relationship between Internet domains and their subdomains, by using a dot-separated syntax. This *LispPlan* delegation process to direct subordinate MESs continues recursively, traversing the

tree structure of all relevant DMES instances. Figure 7 illustrates just one particular structure of DMES instances, but other real system topologies can require different branching of this tree structure of DMES instances, assignments of devices to the MES instances, as well number of the instances. The specific structure depicted in Figure 7 covers two levels of DMES instance subordination and each instance may have 0–$n$ a number of subordinate MESes. Each DMES instance may have 0–$n$ devices assigned to it. Assigning no device to some DMES instance may seem strange at first glance, but may, for example, represent a situation when DMES instances 3 and 4 are spatially distributed from the rest of the system and they work in parallel and should be orchestrated as a whole. This is frequently the case in material handling where two parallel loading or unloading stations operate independently of each other but are managed as a unit by a common fleet or logistics management system. An important observation from Figure 7 is not only the tree structure of DMES instances and their communication with direct parental and child nodes only, but also communication of all DMES instances with one common/shared digital twin that serves as the main entity keeping the actual state of the production system.

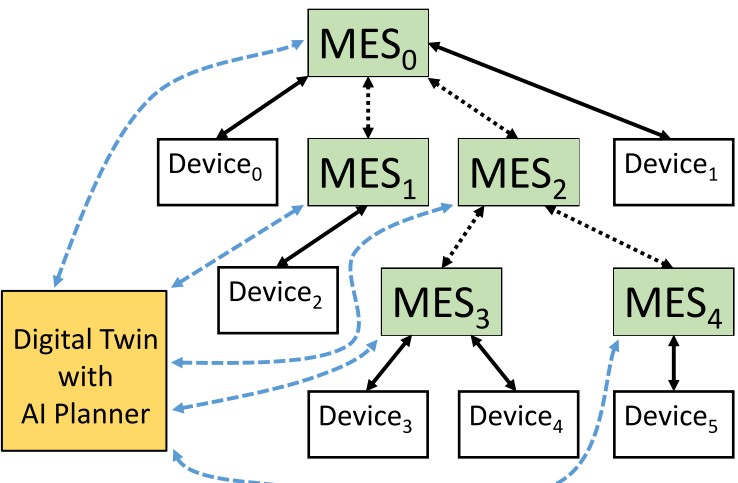

**Figure 7.** A generic example of the hierarchical DMES communication structure. Each DMES instance (denoted $MES_0$–$MES_4$) can control directly connected I40 devices (denoted $Device_0$–$Device_5$) that are spatially and logically close to the specific DMES instance. In order to synchronize the manufacturing execution process, each DMES instance communicates only with its parental DMES instance, or with its subordinate DMES instances, but it cannot communicate with any other instances. In addition, each MES instance communicates with the digital twin to get and to update the state of the production system and its digital replica. The numbers of DMES instances as well as connected devices are arbitrary and set up by system integrators/engineers based on the size and nature of the production system.

A detailed description of this delegation and execution process of the *LispPlan* is given in Algorithm 1: each DMES instance is represented by one running *MES* procedure defined in Algorithm 1. The procedure starts by waiting for the input parameters *plan* and *source* (line 2), where *plan* is a LispPlan and *source* is either a location of the parental MES (i.e., the MES sending the delegated segment of the LispPlan) or `None`, if it is the main DMES instance. After the initialization phase (lines 3–8), the *mes_pool* variable will contain all MESes that are directly connected (for *LispPlan* delegation and execution) to the current MES and are involved in the current *LispPlan* execution. The delegation process of sending sub-plans to directly subordinate MES instances is performed on lines 5–8. Once all parts of the LispPlan that can be delegated to other MES instances have been delegated, the execution core of MES procedure can begin (lines 9–42).

---

**Algorithm 1:** Definition of an instance of a distributed MES

---

1  **procedure** *MES*:
2  **wait** until this MES instance is delegated with a tuple (*plan*: LispPlan, *source*[a]: location of where the *plan* came from) ;
3  *mes_pool* = ∅ ;         `// this set will contain only direct children or parent MESs`
4  **if** *source* is not None **then** add the parent MES from *source* to *mes_pool* ;
5  **foreach** *task*[b] **in** *plan* **using** DFS or BFS ordering[c] **do**
6     **if** *task* is not yet delegated **and** there is another ready/waiting MES instance *m* that is able[d] to process *task* **then**
7         Delegate the *task* and all its *sub-tasks* to MES *m* as a new standalone sub-LispPlan ;
8         Add MES *m* to *mes_pool* ;

9  **foreach** *task*[b] **in** *plan* **where** *task* is not delegated to another MES **do**
10     **if** *task* can be processed by the current *MES* **then**
11         **begin parallel thread**
12             Set the state of *task* to QUEUED;
13             **wait** until all *task* requirements are fulfilled[e] ;
14             Set the state of *task* to PROCESSING ;
15             **if** *task* contains an action **then**
16                 **switch** result of *action start* committed to the Digital Twin **do**
17                     **case** the action has already started **do**
18                         **break** ;
19                     **case** successful but a new plan is available **do**
20                         Report that a new plan is available to this *MES* instance ;
21                     **case** successful `// in accordance with the current plan`
22                     **do**
23                         **if** Process the action on the real hardware **and then**
24                         Commit *action done* to the Digital Twin **then**
25                           Set the state of *task* to DONE ;
26                       **else** Set the state of *task* to FAILED ;
27                       **break** ;
28                     **case** failed but a new plan is available **do**
29                         Report that a new plan is available to this *MES* instance ;
30                     **otherwise do** `// failed and no new plan is available`
31                       Set the state of *task* to FAILED ;
32                       **break** ;
33             **else**
34                 Set the state of *task* to PENDING ;
35                 **wait** until all its sub-tasks are processed or any sub-task is failed ;
36                 **if** all sub-tasks were processed successfully **then**
37                     Set the state of *task* to DONE ;
38                 **else** Set the state of *task* to FAILED ;
39         **terminate thread** ;
40     **else** `// task cannot be processed on the current MES infrastructure`
41         Set the state of *task* to FAILED ;
42         **break** ;

43  **repeat**
44     **wait** until *task* state is changed **or** a new plan is available ;
45     **if** a new plan is available **then**
46         **terminate** all local tasks in state QUEUED or PENDING ;
47         Redistribute to MESs in *mes_pool* that the new plan is available ;
48         **if** *source* is None **then**
49             Download the new plan from the Digital Twin into *plan* ;
50             **goto** line 3 ;         `// The main MES instance starts with the new plan.`
51         **else**
52             **goto** line 2 ;         `// The MES instance will wait for a new plan.`
53     **if** *task* state changes[f] **then** sync *plan* and redistribute to MESs in *mes_pool*;
54  **until** *plan* is completed[g] **or** *MES* is interrupted;
55  **if** *MES* is interrupted because of failure or external intervention **then**
56     **terminate** safely all local tasks ;
57  **goto** line 2 ;         `// The MES instance will wait for a new plan.`

---

[a] to distinguish between a main LispPlan (*source*==None) or a redistributed part of that LispPlan on a specific sub-MES.
[b] including all sub-tasks
[c] any parent must be processed prior to its children
[d] determined from *task* location
[e] all tasks specified in requirements are in DONE state and the parent task is in PROCESSING state
[f] sync is received from another MES
[g] all tasks including all sub-tasks in *plan* are in DONE state

Each task of the LispPlan is processed in parallel in a separate thread. First, the task is initialized to the QUEUED state (line 12). If all the requirements of *task* are met, the execution

of this task can begin and its state changes to PROCESSING. Then two cases need to be resolved (*if* condition at line 15):

1.  If *task* contains no action but has sub-tasks, then (lines 34–38) *task* waits for the sub-tasks to be processed (lines 34–35) in the PENDING state, and then if everything succeeds (line 36), *task* is set to DONE. Otherwise, *task* is set to FAILED (line 38).
2.  If *task* contains an action (and has no sub-task, which is the only option according to the *LispPlan* format), then the process continues on the *switch* statement (lines 16–32). The switch control expression contains the response of the digital twin to *action start* with the following cases:

    (a)  The action has already started in the past.
        Then, leave the *switch* statement (lines 17–18).
    (b)  The action started successfully in the digital twin, but a new plan was computed in the digital twin.
        In this case, the action can start on the real production line because it is valid with the digital twin (the PDDL model and the new plan), but the new plan needs to be downloaded into all DMES instances.
        Because of that, this new plan availability is reported to the current instance of DMES (line 20), and then this case continues to the next subsequent case.
    (c)  The action started successfully (and the current plan is still valid).
        The processing on the real production line is started (line 23) and then *action done* is committed to the digital twin (line 24). If a problem occurs, something unexpected must have happened, and the *task* state is set to FAILED (line 26). Otherwise, everything has been successful and *task* is set to DONE (line 25).
    (d)  The action cannot be started (because it does not conform to the model or plan of the digital twin), and a new plan was computed in the digital twin.
        Then, this new plan availability is reported to the current instance of DMES, and then this case continues to the next subsequent case.
    (e)  The action cannot be started because it does not conform to the model of the digital twin, and the current plan is still valid.
        Let the *task* state be set to FAILED (line 31).

Now, the thread has resolved all the cases and can be safely terminated (line 39).

If any *task* state changes, the current DMES instance is immediately informed / synchronized from lines 12, 14, 25, 26, 31, 34, 37, and 41. Such changes are then gradually propagated (line 52) to all DMES instances. If a new plan availability is reported to the current DMES instance (line 44), then all tasks (which were started by this DMES instance) in state QUEUED or PENDING are terminated, and then information about the new plan availability is reported (line 46) to all neighboring DMES instances from *mes_pool*.

Now, if the current DMES instance is the main instance, then the new plan from the digital twin is downloaded, and the DMES instance is reinitialized by jumping to line 3. Otherwise, the current instance is reinitialized by jumping to line 2, which means waiting for a new sub-plan.

As soon as all of the tasks become DONE, then the LispPlan processing is completed successfully (line 53), and all DMES instances start waiting for the next LispPlan job by jumping to line 2. If any of the tasks become FAILED, then the LispPlan execution has to be interrupted (lines 54–55), the root problems resolved, and the *MES* execution procedure can be restarted with a new valid LispPlan.

*3.4. Basic Digital Twin Based on PDDL Model*

In this article, we use nesting of digital twins. The advantages of this approach include reduced risk of programming errors (decomposing the problem into smaller well-defined units simplifies the code, facilitates the creation of unit tests, and thus reduces the overall risk of errors), and it improves the possible horizontal scalability, which is increasingly important, especially when moving to cloud platforms.

The role of the basic digital twin is to check and interpret the actions defined in the PDDL domain against the current PDDL state. The following actions are supported:

- *action check* — Checks whether the action can be executed.
- *action start* — Starts the action if possible. Otherwise, an error code is returned.
- *action done* — Completes the action if possible. Otherwise, an error code is returned.
- *get state* — Returns the current state of the basic digital twin in terms of a PDDL problem.
- *set domain and problem* — Sets up a new PDDL domain and problem.

The main difference of the basic digital twin compared to the classic PDDL interpreter is the support to initiate multiple operations in parallel. Without this support, it would be impossible to validate the execution of a potentially parallel *LispPlan*. However, the basic PDDL standard does not provide such support and its durative extension is not feasible, as discussed above. Therefore, it was necessary to add support for a retrospective parallelization effort, while supporting a simple and safe form of parallelization of actions. The idea of parallelization in our case is based on the analysis of blocking resources. A blocking resource is a special resource type that cannot be used more than once at the same time. We have used PDDL action parameters to indicate blocking resources for each action. More precisely, if an action contains a parameter with a type derived from `BLOCKING-RESOURCE` type, then this parameter represents a blocking resource.

A simple example of the use of this type mechanism is the PDDL domain description in Figure 1, where all the parameters of both `SHUTTLE_DEPART` and `SHUTTLE_ARRIVE` actions are derived from the `BLOCKING-RESOURCE` type. Note, that the PDDL code in this figure is very simplified in comparison to the full version that is used in our digital twin: e.g., (i) it does not include the locking mechanism that is available in some of the stations for precise positioning and (ii) in the simplification, all the specified types are actually blocking resources which is not the case in the full PDDL domain specification, which cannot be reasonably included, as it would be too large.

When the action starts, the digital twin checks whether all blocking resources of the action are available and whether the precondition of the action is met. All blocking resources of such an action are released only after the action is successfully completed. This mechanism allows multiple actions to run in parallel, and the domain programmer only needs to specify blocking resources for each action in the parameters. Practical experiments have shown that this way of defining parallel actions simplifies and shortens the entire formal domain specification in PDDL. Moreover, any standard off-the-shelf sequential PDDL planner, such as fast downward planning system [78], can be used.

### 3.5. Digital Twin with AI Planner

To extend the potential of digital twin technology, we decided to add more advanced features such as AI planning and scheduling to the external/outer digital twin, and we encapsulated the basic digital twin within it as shown in Figure 8.

The main motivation for this extension of such a digital twin is the idea that both the production goal (i.e., what is to be produced) and the production plan (i.e., how it is to be produced) are actually implicit and explicit parts of the real production line. So, we decided to also reflect this fact (production goal and production plan) in the digital twin as parameters. Since changing the production goal or changing the parameters/resources/skills of the production line means the need to find a new production plan (*LispPlan* in our case), we have integrated an AI planner and scheduler into this digital twin, which can be utilized whenever needed.

## Digital Twin with AI Planner

**Figure 8.** A detailed look at the architecture of a digital twin with the AI planner described in Section 3.5, which includes a nested basic digital twin based on the PDDL production line model (a detailed example is shown in Figure 1) and an AI-powered planner and scheduler. There is also a component for collecting and processing information about the runtime parameters of each action, such as the duration of the action, the energy required to execute the action, etc., and a component for enriching *LispPlan* with metadata of expected energy consumption, duration, etc. An overview of the entire distributed MES architecture, utilizing and benefiting from the digital twin with the AI planner, is given in Figure 4.

An important advantage of the integration of planning into the digital twin is the possibility of deeper semantic control during testing or virtual execution of a production operation/action. Now, the digital twin can detect not only whether a production operation is valid against the PDDL model of the production line (the functionality of the basic digital twin) but also whether the production operation is in accordance with the existing production plan. If it is not in accordance, a new production plan can be automatically replanned. The following operations extend the functionality of the basic digital twin:

- *action check*—Checks if the action can be executed and also if the action is in accordance with the current plan.
- *action start*—Starts the action if possible and reports back whether the action is in accordance with the current plan (if not, a replanning is automatically triggered). Otherwise, an error code is returned.
- *action done*—Completes the action, if possible, and reports back whether the action is in accordance with the current plan (if not, a replanning is automatically triggered). Otherwise, an error code is returned. This action/operation can be enriched with various data measured from the real production line, such as operation time or power consumption.
- *set goal*—Sets a production goal in PDDL format and automatically starts the planning process.

- *set domain and problem*—Sets up a new PDDL domain and problem and then automatically starts the planning process if the goal is already specified.
- *get state*—Returns the current state of the digital twin as a PDDL problem.
- *get plan*—Returns the current *LispPlan*, if any, or reports the status of the planning process. Operations already performed are continuously reflected in the returned plan. A newly computed plan is identifiable by changing the unique hash tag of the plan (calculated as SHA256), which is stored in the plan's metadata. The resulting *LispPlan* also contains in its metadata estimations of the remaining time, the total number of actions/operations, and the total energy consumption.

Another important feature of the digital twin with an AI planner is the ability to improve an existing plan (even while production is running) by speculatively replanning, using a different PDDL planner or the same PDDL planner with a different search strategy.

After a PDDL problem change (i.e., due to a new production goal or a production line state change), the PDDL planner is first run with a search strategy so that the potential resulting plan is found in the shortest possible time, regardless of the quality of the plan. Once such a "quick" plan is found and scheduled, production can immediately begin in the spirit of "time is money". However, since the production process itself takes time, this time can be used for a more time-consuming search for higher quality plans (with shorter execution time, lower energy consumption, or other relevant metrics).

In our experiments we used the fast-downward (lavailable online: https://www.fast-downward.org, accessed on 11 January 2023) PDDL planner. With appropriately optimized search strategy parameters and partial decomposition of the PDDL goal into sub-goals, we were able to find even very complex plans within seconds (Figure 9). On the other hand, finding high-quality plans without decomposing the PDDL problem into sub-problems took easily tens of hours of planning with very high RAM consumption. Given the large variance in the computing times of different planners (or the same planners with different search strategy configurations), we started with the fastest planning strategy and then gradually called potentially higher quality but slower planners if there was still time for further planning. If any of the previously invoked strategies was *complete* (such as A\* [79]) and if no plan was found, we could terminate further searches with the assumption that the *completeness* of the search strategy guaranteed the non-existence of a plan.

There is another important aspect of finding a new plan while an existing plan is being already executed by the DMES. It is the selection of the correct initial state (from all reachable states of the current plan) as a starting point from which the new plan will be computed. In case the currently executed plan reaches or even exceeds the assumed initial state while the new plan has not yet been found, the search for this new plan should be canceled, as it would not be possible to continue after its eventual completion because the current plan and the new plan no longer share a common state from which DMES could switch from one plan to the other. The following different strategies/approaches can be considered to select a suitable starting point:

1. *Empirical ad-hoc* approach, where each search strategy has its own time limit based on the empirical knowledge of the programmer. If this time limit is exhausted, the search stops and another search strategy is used instead. The pre-known time limit for each search strategy determines the appropriate starting point for beginning the planning process.
2. *Backward iterative deepening* approach, where planning starts at the point of the last operation before the end of the current plan and then iteratively extends in time toward the beginning of the plan. Since the computation time of such a plan usually increases exponentially with the expected length of the plan, the time required to compute all previous plans usually does not significantly exceed the computation time of the new plan in the next iteration.
3. *Machine learning estimation* approach where the task is to estimate the duration of the planning process. Inputs can be a suitable representation of the particular search strategy used, the PDDL goal, and possibly even the entire PDDL problem.

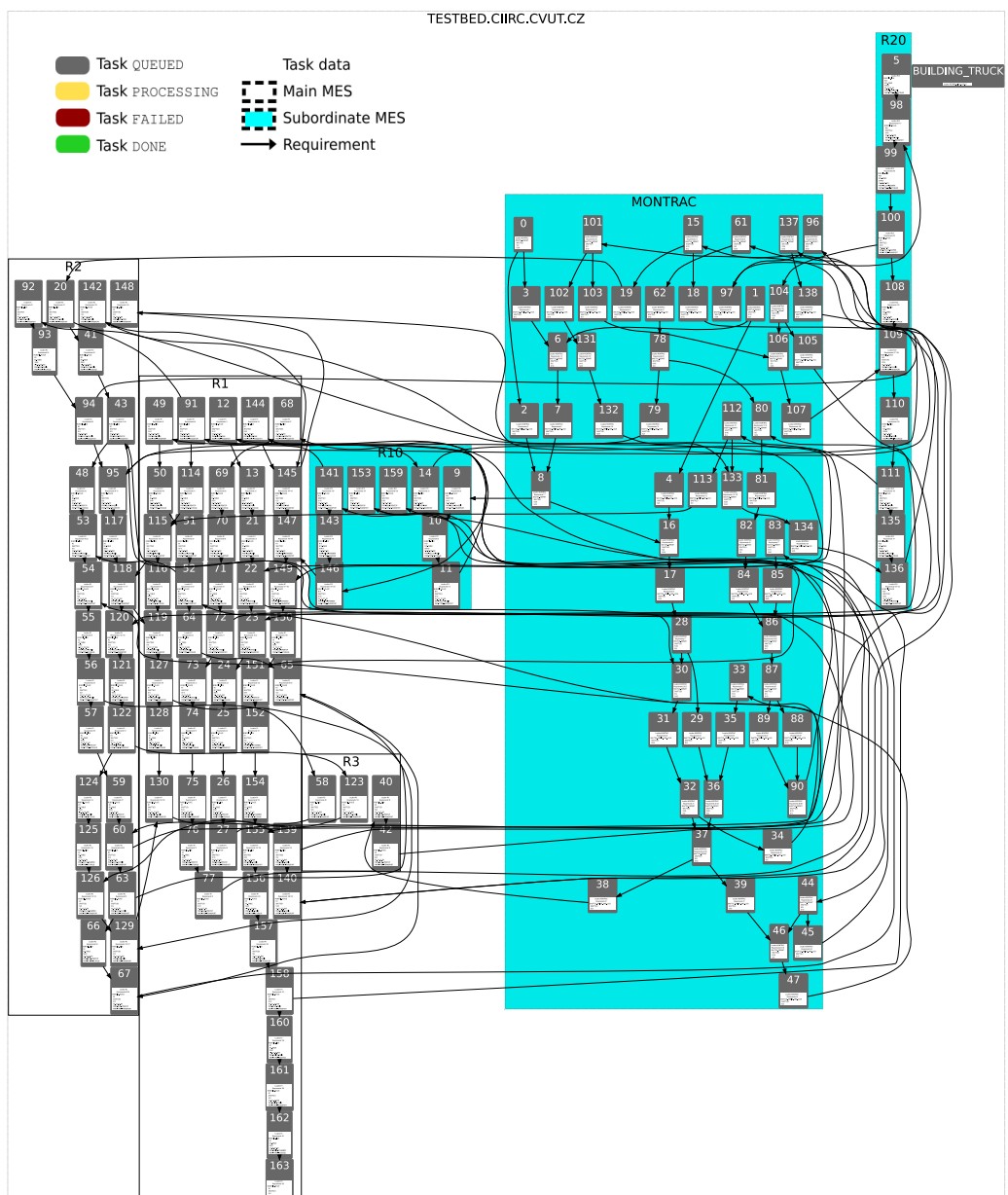

**Figure 9.** An example of a rather complex *LispPlan*, which was automatically computed using the digital twin with AI planner in less than ten seconds. It consists of 164 operations and produces three different trucks at once. The purpose of this figure is not to go into detail, but to illustrate the manageable complexity of the proposed on-the-fly planning in conjunction with flexible manufacturing execution.

In this article, we have implemented the first approach, but we plan to focus on the other two in future research. Of course, several search strategies could be run in parallel, but in this article we relied on a sequential approach to run different search strategies. However, we are planning to parameterize the maximum number of parallel running search instances inside the digital twin with AI planner to achieve (i) better scalability and (ii) an overall speed-up of the entire planning and, subsequently, the production process.

## 4. Evaluation and Discussion

Our implementation was evaluated on the I40 testbed, based on the assumption that the use of our digital twin with an AI planner with its online support for instantaneously changing the executed plan in our distributed MES can effectively reduce the energy consumption of production, and thus contribute to better sustainability in industry.

The following example illustrates the overall process of our DMES together with the digital twin with AI planner. It all starts with a production order for two trucks (truck one consists of a black chassis, a blue cabin, and a white stakebed and it will be finally transported to station S200; truck two consists of a black chassis, a white cabin, and a yellow dumper and will be finally transported to station S100). It is transmitted in the following PDDL goal formulation:

```
(:goal
  (and
    (exists
      (?P1 - POSITION ?P2 - POSITION ?P3 - POSITION
       ?S1 - SHUTTLE-RESOURCE ?S2 - SHUTTLE-RESOURCE)
      (and
        (RESOURCE_CONTAINS_PART_AT ?S1 PART-TYPE-T-TRANSPORT PART-BLACK-CHASSIS ?P1)
        (RESOURCE_CONTAINS_PART_AT ?S1 PART-TYPE-T-TRANSPORT PART-YELLOW-DUMPER ?P2)
        (RESOURCE_CONTAINS_PART_AT ?S1 PART-TYPE-T-TRANSPORT PART-WHITE-CABIN ?P3)
        (IS_RESOURCE_ENABLED ?S1)
        (RESOURCE_CONTAINS_PART_AT ?S2 PART-TYPE-T-TRANSPORT PART-BLACK-CHASSIS ?P1)
        (RESOURCE_CONTAINS_PART_AT ?S2 PART-TYPE-T-TRANSPORT PART-WHITE-STAKEBED ?P2)
        (RESOURCE_CONTAINS_PART_AT ?S2 PART-TYPE-T-TRANSPORT PART-BLUE-CABIN ?P3)
        (IS_RESOURCE_ENABLED ?S2)
        (RESOURCE_IN_STATION ?S1 S200)
        (RESOURCE_IN_STATION ?S2 S100))))
)
```

This PDDL goal is then sent to the digital twin with an AI planner. There, the current production state is retrieved from the basic digital twin, as well as the expected action costs from the measured runtime data processor and collector. From these data, the problem generator creates the PDDL domain and the problem for the PDDL AI planner. If a solution is found in the form of a sequence of actions, then it is sent to the scheduler to be parallelized in terms of a *LispPlan*. Next, the *LispPlan* parameter estimator enriches with the following additional metadata (hash ID of the plan and calculated estimates of energy consumption, plan execution duration, etc.):

```
(:METADATA
            (:ID
                C999DCD2A01F6B0902055CDAF5F0A05D93F7A80C98ED0282AC9539F7C77E5D80
            )
            (:DURATION-IN-SECONDS 360.34093340513164)
            (:TOTAL-TIME-SPENT-IN-SECONDS 490.91739162084514)
            (:ENERGY-IN-WATT-HOURS 13.051901711931999)
            (:TOTAL-ACTIONS 20)
)
```

Table 1 provides the expected execution time and energy consumption of operations that have already been collected and processed in previous runs and that are used for the estimates in the metadata. Next, *LispPlan* is sent back to the main instance of the DMES, where it is partitioned for each MES instance. The resulting plan visualization (including partitioning) from this stage can be seen in Figure 10. In this particular case, three MES instances were used:

- The main `TESTBED.CIIRC.CVUT.CZ` MES instance (marked by the outer dashed line with transparent fill in Figure 10).
- The `MONTRAC.TESTBED.CIIRC.CVUT.CZ` MES instance (marked by a dashed line with cyan fill in Figure 10).
- The `R20.TESTBED.CIIRC.CVUT.CZ` MES instance (marked by a dashed line with cyan fill in Figure 10).

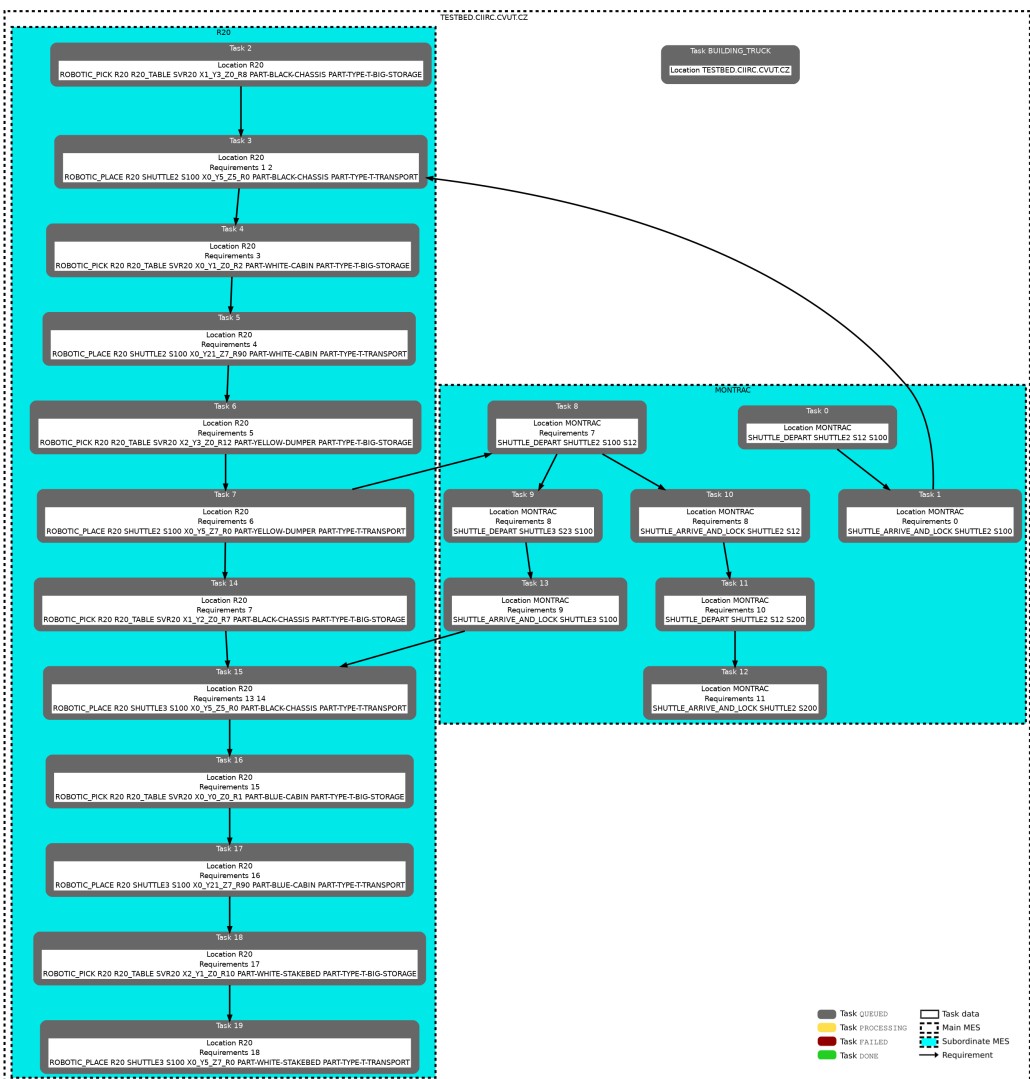

**Figure 10.** An example of a *LispPlan* that was computed in less than one second using the fastest heuristic (with worst quality) composed of several `fast-downward` calls on PDDL subgoals. The visualization is part of the DMES web interface and was automatically generated from the *LispPlan*.

The entire process from the beginning up to this point took less than one second of computation time on a single core of the I40 testbed server (Intel® Xeon® CPU E5-2630 v2 @ 2.6 GHz, 4 cores in total, 32 GB RAM, 500 GB SSD storage, OS: Ubuntu 20.04.4 LTS (GNU/Linux 5.4.0-121-generic x86_64)).

At this point, the real production as well as the speculative replanning inside the digital twin has started. More precisely, the first two actions without requirements (Task 0 and Task 2) were started on the digital twin (by calling *action start*) and on the real production line.

While both actions are being executed and later passed to the digital twin (by calling *action done*), a new and better plan was found. The information about the new plan is now propagated to the DMES and subsequently distributed to all MES instances as described in Algorithm 1. The situation, where a new action has already started with the new plan, is visualized in Figure 5. The resulting *LispPlan* source code of the new plan is shown in Figure 6.

The replanning resulted in saving two actions that performed shuttle movement. Task 8: `SHUTTLE_DEPART` and Task 10: `SHUTTLE_ARRIVE_AND_LOCK` of the old plan are not necessary in the new plan. While the total estimated time to complete the plan (`:DURATION-IN-SECONDS`) has remained almost the same, the expected energy savings due to the reduction in the number of operations sum up to almost 0.87 Wh. In reality, the energy savings were lower than expected (approximately 0.77 Wh) because our model for

estimating the energy consumption of individual operations is not able to simulate complete behavior of measured devices, especially when measuring the operations performed by the Montrac transport system (so we now estimate the consumption proportionally to the duration of the operation only). Table 2 reports all measurements of execution times and energy consumption made during the production of this example on an I40 testbed production line.

**Table 1.** Selected measured execution time and energy consumption of production operations related only to the use-case in Section 4 out of a total of 1113 recorded operations at I40 testbed. The symbol "..." indicates that the parameter (or part of it) in operation is not relevant for the measurement and is therefore not included in the table. Every measured operation in this table was recorded at least ten times. The energy measurement is the difference between the energy consumption in the idle state and the energy consumption in the active state on a specific related device.

| Operation/Action | Execution Time (Mean ± SD) [s] | Energy Consumption (Mean ± SD) [Wh] |
|---|---|---|
| (ROBOTIC_PICK R20 ...  SVR20 ...R8 PART-BLACK-CHASSIS ...) | 16.74 ± 0.17 | 0.9626 ± 0.0099 |
| (ROBOTIC_PLACE R20 ...  S100 ...  PART-BLACK-CHASSIS ...) | 10.73 ± 0.26 | 0.3291 ± 0.0079 |
| (ROBOTIC_PICK R20 ...  SVR20 ...  PART-WHITE-CABIN ...) | 15.84 ± 0.23 | 0.9106 ± 0.0130 |
| (ROBOTIC_PLACE R20 ...  S100 ...  PART-WHITE-CABIN ...) | 12.36 ± 0.20 | 0.3789 ± 0.0062 |
| (ROBOTIC_PICK R20 ...  SVR20 ...  PART-YELLOW-DUMPER ...) | 17.00 ± 0.23 | 0.9776 ± 0.0131 |
| (ROBOTIC_PLACE R20 ...  S100 ...  PART-YELLOW-DUMPER ...) | 10.53 ± 0.12 | 0.3228 ± 0.0037 |
| (ROBOTIC_PICK R20 ...  SVR20 ...R5 PART-BLACK-CHASSIS ...) | 15.73 ± 0.03 | 0.9045 ± 0.0017 |
| (ROBOTIC_PICK R20 ...  SVR20 ...  PART-BLUE-CABIN ...) | 15.48 ± 0.28 | 0.8903 ± 0.0159 |
| (ROBOTIC_PLACE R20 ...  S100 ...  PART-BLUE-CABIN ...) | 12.39 ± 0.17 | 0.3799 ± 0.0052 |
| (ROBOTIC_PICK R20 ...  SVR20 ...  PART-WHITE-STAKEBED ...) | 16.00 ± 0.38 | 0.9202 ± 0.0219 |
| (ROBOTIC_PLACE R20 ...  S100 ...  PART-WHITE-STAKEBED ...) | 10.45 ± 0.04 | 0.3203 ± 0.0013 |
| (SHUTTLE_DEPART SHUTTLE2 S12 S100) | 0.52 ± 0.36 | 0.0017 ± 0.0012 |
| (SHUTTLE_ARRIVE_AND_LOCK SHUTTLE2 S100) from S12 | 117.40 ± 6.36 | 1.9566 ± 0.1060 |
| (SHUTTLE_DEPART SHUTTLE3 S100 S200) | 0.49 ± 0.21 | 0.0016 ± 0.0007 |
| (SHUTTLE_ARRIVE_AND_LOCK SHUTTLE3 S200) from S100 | 50.67 ± 1.91 | 0.8446 ± 0.0318 |
| (SHUTTLE_DEPART SHUTTLE2 S12 S200) | 0.51 ± 0.05 | 0.0017 ± 0.0002 |
| (SHUTTLE_ARRIVE_AND_LOCK SHUTTLE2 S200) from S12 | 61.91 ± 1.20 | 1.0319 ± 0.0200 |
| (SHUTTLE_DEPART SHUTTLE3 S23 S100) | 0.57 ± 0.19 | 0.0019 ± 0.0006 |
| (SHUTTLE_ARRIVE_AND_LOCK SHUTTLE3 S100) from S23 | 109.74 ± 3.13 | 1.8289 ± 0.0521 |

Finally, due to the design of the proposed DMES and the digital twin with AI planner, it was possible to successfully switch from the old plan to the new one (according to case (b) in the description of Algorithm 1).

On the one hand, this illustrative example showed real energy savings in the execution of two fewer actions/operations, representing more than 5%, as a result of the new planning compared to the first quickly found plan. On the other hand, if we run the better planning strategy (parameters are described in Figure 6) immediately on the original problem, it would need 30 seconds of computation with the same resulting quality as the new plan, which applied two strategies that need less than one second of computation before production can start. Given that the idle power consumption of the entire production line of the I40 testbed is at least 410 W and the peak power consumption is not more than 1100 W, this means that this illustrative example showed real power saving of at least 3% due to the production with the new planning compared to the production with the plan found by the better strategy only. Based on the test scenarios already performed during long-term testing in the Industry 4.0 testbed and on our previous results [41], where the only criterion was the overall plan execution time, a reduction in the total execution time of the plan has been achieved by reducing the number of operations/actions used. This leads to energy savings in almost all cases. Therefore, we are convinced that the presented DMES with integrated AI-based planning can achieve up to 30% energy savings compared to the first/initial plan found (i.e., using an aggressive search strategy to reach the PDDL goal as quickly as possible). For further data on execution time reduction using continuous replanning, see our previous article [33].

**Table 2.** Measured execution time and energy consumption of production operations during a particular run of the use-case in Section 4. The symbol "..." indicates that the parameter (or part of it) in operation is not relevant for the measurement and is, therefore, not included in the table. The asterisk next to the task ID (two upmost rows of the table) indicates the task/operation from the first/fastest plan, shown in Figure 10. The operation IDs without the asterisk refer to the improved plan, depicted in Figure 5. The energy measurement is the difference between the energy consumption in the idle state and the energy consumption in the active state of specific related devices.

| Task ID | Operation/Action | Execution Time [s] | Energy Consumption [Wh] |
|---|---|---|---|
| 0* | (SHUTTLE_DEPART SHUTTLE2 S12 S100) | 0.65 | 0.0022 |
| 2* | (ROBOTIC_PICK R20 ... SVR20 ...R8 PART-BLACK-CHASSIS ...) | 16.67 | 0.9586 |
| 0 | (SHUTTLE_ARRIVE_AND_LOCK SHUTTLE2 S100) from S12 | 121.33 | 2.0221 |
| 1 | (ROBOTIC_PLACE R20 ... S100 ... PART-BLACK-CHASSIS ...) | 10.68 | 0.3276 |
| 2 | (ROBOTIC_PICK R20 ... SVR20 ... PART-WHITE-CABIN ...) | 15.85 | 0.9111 |
| 3 | (ROBOTIC_PLACE R20 ... S100 ... PART-WHITE-CABIN ...) | 12.39 | 0.3799 |
| 4 | (ROBOTIC_PICK R20 ... SVR20 ... PART-YELLOW-DUMPER ...) | 17.16 | 0.9870 |
| 5 | (ROBOTIC_PLACE R20 ... S100 ... PART-YELLOW-DUMPER ...) | 10.65 | 0.3265 |
| 6 | (ROBOTIC_PICK R20 ... SVR20 ...R5 PART-BLACK-CHASSIS ...) | 16.39 | 0.9423 |
| 7 | (SHUTTLE_DEPART SHUTTLE3 S100 S200) | 0.50 | 0.0017 |
| 8 | (SHUTTLE_DEPART SHUTTLE3 S23 S100) | 0.51 | 0.0017 |
| 9 | (SHUTTLE_ARRIVE_AND_LOCK SHUTTLE3 S100) from S23 | 109.74 | 1.8289 |
| 10 | (ROBOTIC_PLACE R20 ... S100 ... PART-BLACK-CHASSIS ...) | 10.58 | 0.3245 |
| 11 | (ROBOTIC_PICK R20 ... SVR20 ... PART-BLUE-CABIN ...) | 15.31 | 0.8806 |
| 12 | (ROBOTIC_PLACE R20 ... S100 ... PART-BLUE-CABIN ...) | 12.39 | 0.3798 |
| 13 | (SHUTTLE_ARRIVE_AND_LOCK SHUTTLE3 S200) from S100 | 51.22 | 0.8536 |
| 14 | (ROBOTIC_PICK R20 ... SVR20 ... PART-WHITE-STAKEBED ...) | 15.96 | 0.9175 |
| 15 | (ROBOTIC_PLACE R20 ... S100 ... PART-WHITE-STAKEBED ...) | 10.60 | 0.3252 |
| | Sum total | 447.92 | 12.3687 |

## 5. Conclusions and Future Work

Decreasing the size of production lots in production systems requires a very high degree of flexibility in production planning and execution, along with a suitable product design and production system setup. This article addresses the automation system solution that enables on-the-fly planning and replanning in such a way that the production of a required product can be started with an initial version of an automatically generated production plan. The main idea behind the initial plan is that it can be found quickly by the AI planner at the cost of a possible loss of quality. Once this initial plan is being executed, it can be improved by replanning using different potentially more computationally intensive strategies, leading to a higher quality plan with respect to production metrics such as energy efficiency, total production time, or resources used. Since the newly found plan must take into account the operations already started (where they may have been started while searching for this plan), it is essential that the planner is tightly connected to the digital twin to guarantee a collision-free transition from the original plan to the new one. Encapsulating the digital twin (basic digital twin based on PDDL model) with a planner into a new enhanced digital twin (digital twin with AI planner) is one of the most essential cornerstones of this article.

The proposed solution is in direct contrast to current industrial solutions that are based on predefined production recipes that are hard-coded or embedded in manufacturing execution systems, as well as industrial devices used in daily industrial practice right now. The proposed solution combines the advantage of replanning capabilities with a distributed MES. This approach enables, for example, a reaction to erroneous states instantly and a replanning of the production on-the-fly. Therefore, the proposed solution targets the emerging generation of true Industry 4.0 systems capable of high flexibility and reconfigurability. This realization has been supported by advances in AI methods on the level of planners, but also on the level of justifying the running production plans, mimicking skills of human practitioners, and investigating newly emerging orchestration and integration patterns for industrial automation.

Testing and validation of the proposed approach has been done in CIIRC's I40 testbed during long-term extensive validation studies and presentations for scientists, industrial

practitioners, and even to a wide public audience. The prototype implemented during this validation period showed that this solution is not just a vision of the far future, but it is a viable and fully functional option, tested on a laboratory scale. This testing period proved that the proposed solution is appropriate for long-term and robust deployment.

The benefits of the proposed digital twin-centric solution for flexible and sustainable manufacturing, i.e., (i) flexible reactions to errors and changes in production, together with (ii) flexible production following the current needs in the volatile market, lead to a better utilization of resources and facilitate risk management for I40 production. The approach is applicable even in incrementally designed I40 systems and processes, since its realization is based on a software solution and does not require changing machines or other hardware components (if the shop-floor devices have communication capabilities and can be flexibly controlled). The approach also improves technical and organizational interoperability in the automation verticals within smart factories. The level of flexibility fosters resilience and reliability of the entire manufacturing system and its automation.

*Future Work*

In future work, it would be interesting to test the proposed approach on a real production facility in order to perform stress tests under large-scale industrial conditions. In our experience, real industrial usage brings significantly higher amounts of data to be processed, and it would be worthwhile to check response times under such conditions.

Additionally, investigating the distribution of the digital twin with separate models coupled with planners in a similar manner, like the distributed MES instances, could prove useful. The intention would be improving the level of reuse and reconfigurability in large-scale production system engineering projects, in order to be able to easily and reliably reinstantiate selected parts of production lines to other applications, such as to duplicate one production line to a new one. Such operations seem to be simple at first, but they frequently require significant rework with a risk for numerous mismatches (e.g., in IP addresses). The goal should be to eliminate manual rework and to efficiently support human engineers with a model-based implementation for such engineering work.

**Author Contributions:** J.V., P.D. and P.N. contributed to the main idea and to the conceptualization. The software prototypes were implemented by J.V. and P.D. The validation and testing was done by J.V. and P.D. The structure of the article was designed by J.V. The whole article was written, finalized and proofread by J.V., P.D., P.N. and B.W. All authors have read and agreed to the published version of the manuscript.

**Funding:** This work was funded by the Ministry of Education, Youth and Sport of the Czech Republic within the project Cluster 4.0 (No. CZ.02.1.01/0.0/0.0/16_026/0008432) and the RICAIP project that has received funding from the European Union's Horizon 2020 research and innovation programme under grant agreement No. 857306. The publication was created with support of the project "Regeneration of used batteries from Electric Vehicles" (Slovak ITMS2014+ code 313012BUN5), which is a part of the Important Project of Common European Interest (IPCEI) called the European Battery Innovation (code OPII-MH/DP/2021/9.5-34), announced as a part of Operational Program Integrated Infrastructure (EZOP ID 71235).

**Institutional Review Board Statement:** Not applicable.

**Informed Consent Statement:** Not applicable.

**Data Availability Statement:** Examples of data from this study can be made available upon request.

**Conflicts of Interest:** The authors declare no conflict of interest.

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
