# Peer review of "A Digital Twin-Based Distributed Manufacturing Execution System for Industry 4.0 with AI-Powered On-The-Fly Replanning Capabilities"

_sustainability, doi:10.3390/su15076251_

Round 1

Reviewer 1 Report

The paper aimed to propose a new MES architecture based on standalone instances for Industry 4.0, employing digital twin and artificial intelligence planning methods to compose, verify, and interpret a production execution plan autonomously. The authors tested the proposal architecture on a setup simulating a shop floor production line, resulting in a reduction of energy consumption through the good performance of the AI planner. 

The paper is well-written and well-organized, and the content is described clearly and consistently. In addition, the authors provide a good background and explain future works.

1.   The introduction section's first paragraph is missing the references for the statements.

2.   Figure 6 is not clear. Even the legends are not readable. The figure is briefly mentioned in the text to give an idea of a complex system. I suggest removing or showing a small part of the system that can be readable.

3. Is there an estimation of the reduced energy consumption percentage during the operation?

Author Response

Reviewer’s comment 1:
The introduction section's first paragraph is missing the references for the statements.

Authors’ response 1:
We have added further references to support the statements in the Introduction. In addition, we have improved and polished the whole argumentation line in the Abstract and Introduction section.

Reviewer’s comment 2:
Figure 6 is not clear. Even the legends are not readable. The figure is briefly mentioned in the text to give an idea of a complex system. I suggest removing or showing a small part of the system that can be readable.

Authors’ response 2:
We have regenerated the figure to make it more readable. We have also added a better explanation that this figure shows rather a feasible complexity of tasks/problems that can be planned and re-planned on-the-fly. Therefore, it is not important to read all labels, but to get an idea how such a plan in the form of a graph can look like for more complex cases than are educatory or illustrative demonstrations.

Reviewer’s comment 3:
Is there an estimation of the reduced energy consumption percentage during the operation?

Authors’ response 3:
Based on the test scenarios performed during long-term testing within the Industry 4.0 Testbed and based on our previous results, we have experienced that we can reach energy saving up to 30 percent compared to the first/initial found plan (that is using aggressive fast search strategy).
We have also added this observation to the manuscript, that is, in the last paragraph of Section 4.

Reviewer 2 Report

1- Abstract

A brief summary of the research's purpose is provided.

However, the initial brief statement of the issue is excessively lengthy.

I would advise adding the theoretical or practical contribution in the work's abstract.

2- Introduction

It is advised that the authors lay out the article's overall structure in the last paragraph.

3- Materials and Methods

In this section, the authors presented background and foundations for describing and understanding the proposed planning and scheduling of industrial production systems controlled by a distributed MES. Why was it placed in the "materials and methods" section rather than the "literature" section?

4- Implementation and Results

Given that the authors have used the first approach and intended to concentrate on the other two in subsequent research, I would recommend including the research flow diagram (in the Methodology section) because it is very interesting and exciting.

Author Response

Reviewer’s comment 1:
1- Abstract
A brief summary of the research's purpose is provided. However, the initial brief statement of the issue is excessively lengthy. I would advise adding the theoretical or practical contribution in the work's abstract.

Authors’ response 1:
We have made a number of changes to the abstract and we believe it is now easier to read and understand.
Specifically, we have revised the abstract by removing some of the original lengthy statements, and we have added a better summary of the contribution of this work.

Reviewer’s comment 2:
2- Introduction
It is advised that the authors lay out the article's overall structure in the last paragraph.

Authors’ response 2:
The layout of the overall structure of the work has been added as the last paragraph in the Introduction section.

Reviewer’s comment 3:
3- Materials and Methods
In this section, the authors presented background and foundations for describing and understanding the proposed planning and scheduling of industrial production systems controlled by a distributed MES. Why was it placed in the "materials and methods" section rather than the "literature" section?

Authors’ response 3:
We have used the MDPI recommended template, which in our understanding recommends calling the section summarizing the literature/state-of-the-art 'Materials and Methods'. Although even we agree with your opinion, we have decided to keep this original and recommended section name.

Reviewer’s comment 4:
4- Implementation and Results
Given that the authors have used the first approach and intended to concentrate on the other two in subsequent research, I would recommend including the research flow diagram (in the Methodology section) because it is very interesting and exciting.

Authors’ response 4:
In our opinion, Section 3.5, which describes the digital twin with the AI planner and contains three proposed methods to implement the time estimation of the planning process, is only a support to explain the entire problem of integrating the digital twin with the AI planner. Our intention is to focus on this research question in the next dedicated paper. Therefore, we have not added a dedicated research methodology section to this manuscript.

Reviewer 3 Report

Authors should double-check the flow of the article. English and grammar are still issue! This reviewer sees that ‘system’ is repeated in so many places starting the first sentence of the paper. Major re-writing is required.

The topic presented is very hot today’s R&D. The references and the depth provided in Introduction is weak. More articles and depth are needed.

After reading the first two pages, this reviewer believes that the title is a mess and does not represent the coverage of the paper. It needs to be re-written.

Majority of the figures presenting the system architecture has no meaning. It is unclear and hard to understand.

A generic graphics is required to show the overall structure of the paper’s goal. It is still hard to figure out the codes and layouts presented.

Figure 1 was not labeled and detailed. It means just a robot. What else?

Figure 3 and Figure 4 components were not detailed and explained.

What is the reasoning to show the codes inside the research paper? They should be taken out or presented inside the appendix if they are definitely needed.

In evaluation, there is no measurable data or evidence. How do you evaluate and prove that evaluation was a success?

The digital twin concept was used more than 80 times in this paper. However, the authors were not able to draw a tangible picture on their digital twin system clearly.

Author Response

Reviewer’s comment 1:
Authors should double-check the flow of the article. English and grammar are still issue! This reviewer sees that ‘system’ is repeated in so many places starting the first sentence of the paper. Major re-writing is required.

Authors’ response 1:
We have significantly updated and improved the argumentation line and flow of the article. In addition, we have improved the quality of the text, English language, and polished the entire manuscript. The word 'system' is used in the work in multiple contexts. First, it is a part of the common abbreviation MES (Manufacturing Execution System), which is a type of an automation tool that we are extending with new features. Second, 'system' is a part of a common and well-established terms: 'control system', 'automation system', 'cyber-physical system', 'real-time system', 'transportation system', 'production system engineering'; In our opinion, these terms are not suitable to be rephrased, because a reader can easily be confused.

Reviewer’s comment 2:
The topic presented is very hot today’s R&D. The references and the depth provided in Introduction is weak. More articles and depth are needed.

Authors’ response 2:
We have added additional references to better substantiate our statements. A more in-depth explanation of the related work is described in the 'Materials and Methods' section, which is the recommended title/name of the MDPI Sustainability journal template.

Reviewer’s comment 3:
After reading the first two pages, this reviewer believes that the title is a mess and does not represent the coverage of the paper. It needs to be re-written.

Authors’ response 3:
The title of the article includes the key term “Distributed Manufacturing Execution System” and emphasizes the use of “Digital Twins”. Since our digital twin has been enhanced with an integrated AI planner and scheduler, we have included the formulation “AI-powered On-the-fly Replanning Capabilities”, which in our opinion highlights significant benefits of the proposed solution. We have improved the argumentation line of the manuscript to make it more understandable and clearer.

Reviewer’s comment 4:
Majority of the figures presenting the system architecture has no meaning. It is unclear and hard to understand.

Authors’ response 4:
We have double-checked and improved the description of figures, including their textual references/descriptions. We have also regenerated and improved Fig. 6.

Reviewer’s comment 5:
A generic graphics is required to show the overall structure of the paper’s goal. It is still hard to figure out the codes and layouts presented.

Authors’ response 5:
The overall structure of the work, including its goals, are described in the Introduction section (specifically in paragraphs 5 and 6). 

All the codes presented in the manuscript can be split into the following three cases:
(i) The codes in the text of Sec. 4 and in Fig. 5 are in PDDL as mentioned in the description and basic understanding of Lisp notation is required. 
(ii) The code in Fig. 9 is an example of LispPlan and the corresponding visualization is in Fig. 8. 
(iii) The pseudo-code used in Algorithm 1 is explained in detail in Sec. 3.3.

The layout of the overall structure of the work has been added as the last paragraph in the Introduction section.

Reviewer’s comment 6:
Figure 1 was not labeled and detailed. It means just a robot. What else?

Authors’ response 6:
We have improved the caption of the figure and better emphasized that the robot is gripping a yellow car dumper body in its end-effector when completing an assembly process for one of the two trucks from the use-case described later on. This image is attached as proof that the proposed solution has been verified on the real machinery at Industry 4.0 Testbed located at the Czech Technical University in Prague.

Reviewer’s comment 7:
Figure 3 and Figure 4 components were not detailed and explained.

Authors’ response 7:
Figure 3 presents the architecture of the proposed solution. The figure includes numbers in circles/labels that are addressed in detail in the text (Sec. 3.2 and 3.3). Figure 4 presents an example of MES instances and their communication among other instances, which is hierarchical only, and with the digital twin enhanced with the AI planner, where each MES instance communicates with this enhanced artifact. The figure is described and explained in detail in Sec. 3.3. 

Reviewer’s comment 8:
What is the reasoning to show the codes inside the research paper? They should be taken out or presented inside the appendix if they are definitely needed.

Authors’ response 8:
We have decided to show the pseudo-code (Algorithm 1) to explain our solution in detail. 
Our intention is to ensure the reproducibility of the proposed approach, as this pseudo-code allows readers to reimplement the ideas of the manuscript in any programming language they prefer.
The codes in PDDL (in the text of Sec. 4 and in Fig. 5) and in LispPlan (Fig. 9) are essential to understand the nature of the digital twins and overall solution proposed in this work. All these codes have already been modified and reduced for simplicity and better understanding.

Reviewer’s comment 9:
In evaluation, there is no measurable data or evidence. How do you evaluate and prove that evaluation was a success?

Authors’ response 9:
We have added estimated savings into Sec. 4 - Discussion and Evaluation.
Based on our test scenarios, the AI-powered planning can achieve up to 30 percent of energy savings compared to the first/initial plan found with an aggressive search strategy.
In a specific example (Section 4), we achieved a real energy savings of 19 percent due to the replanning capability integrated into the digital twin and the seamless adaptation of production on-the-fly at the DMES level. 

Reviewer’s comment 10:
The digital twin concept was used more than 80 times in this paper. However, the authors were not able to draw a tangible picture on their digital twin system clearly.

Authors’ response 10:
The most important take-away about our digital twin concept from this manuscript is that we have enhanced our digital twin with a production planner and scheduler, which brings numerous benefits including the possibility of replanning a running production on-the-fly. A brief introduction to digital twin terminology is presented in Sec. 2.3. The basic digital twin based on symbolic PDDL model representation including an example (Fig. 5) is explained in Sec. 3.4. The proposed enhancement of the digital twin with the AI planner is depicted in Fig. 3. Later on, a very detailed insight into realization of this digital twin is provided in Sec. 3.5.

Round 2

Reviewer 3 Report

Authors did not provide an in-depth revision. Authors provided answers and arguments. However, there are no substantial changes and enhancements. Several suggested changes are still the same with the current paper. 

This reviewer suggests a REJECTION decision for this paper since the authors did not consider majority of the revision requests.

Author Response

Reviewer's request:

Authors did not provide an in-depth revision. Authors provided answers and arguments. However, there are no substantial changes and enhancements. Several suggested changes are still the same with the current paper. 

This reviewer suggests a REJECTION decision for this paper since the authors did not consider majority of the revision requests

Authors' response:

  1. We have checked and improved the entire flow of the article, improved English, and overall polished the quality of the manuscript. We have made substantial improvements and enhancements to the whole manuscript.
  2. We have added a significant number of additional references to better substantiate our statements.
  3. We have changed the title of the manuscript to better represent the coverage of the manuscript.
  4. We have double-checked and improved captions of figures, including their textual references/descriptions.
  5. We have added a new figure (namely Fig. 6), depicting a detailed view on our realization of the digital twin in a tangible way. The caption of the figure links other artifacts of our approach, such as PDDL specifications (depicted in Fig. 5) or the whole automation system architecture (depicted in Fig. 3) to make all inter-links in the manuscript more clear.
  6. Regarding the codes included in the manuscript, we feel that it makes sense to have the codes in place in order to provide readers with the possibility of repeatability/reproducibility and the satisfactory level of detail of the proposed approach. The other two reviewers accepted the manuscript with the codes in place, hence, we do not want to change this as it is inline with our opinion but also with the opinion of the other reviewers.